# Out-of-distribution Detection with Test Time Augmentation and Consistency Evaluation

## Abstract

Deep neural networks are known to be vulnerable to unseen data: they may wrongly assign high confidence scores to out-distribution samples. Existing works try to handle the problem using intricate representation learning methods and specific metrics. In this paper, we propose a simple yet effective post-hoc out-of-distribution detection algorithm named Test Time Augmentation Consistency Evaluation (TTACE). It is inspired by a novel observation that in-distribution data enjoy more consistent predictions for its original and augmented versions on a trained network than out-distribution data, which separates in-distribution and out-distribution samples. Experiments on various high-resolution image benchmark datasets demonstrate that TTACE achieves comparable or better detection performance under dataset-vs-dataset anomaly detection settings with a $60\% \sim 90\%$ running time reduction of existing classifier-based algorithms. We also provide empirical verification that the key to TTACE lies in the remaining classes between augmented features, which previous works have partially ignored.

## 1 Introduction

Deep neural networks have shown substantial flexibility and valuable practicality in various tasks (Huang et al., 2017; LeCun et al., 2015). However, when deploying deep learning in reality, one of the most concerning issues is that deep models are overconfident when exposed to unseen data (Amodei et al., 2016; Goodfellow et al., 2015). That is, although the deep models generalize well on unseen datasets drawn from the same distribution (in-distribution data), it incorrectly assigns high confidence to unseen data drawn from another distribution (out-distribution data). To handle the issue, out-of-distribution detection (Aggarwal, 2013) aims to separate out-distribution data from in-distribution data.

Existing out-of-distribution (OOD) research is dedicated to finding and designing a metric that exhibits significant differences in value between in-distribution and out-distribution data. Common methods include tuning a well-trained model (Liang et al., 2018; Lee et al., 2018), training an extra OOD model (Schölkopf et al., 1999; Tack et al., 2020), or some extra OOD functions (Tack et al., 2020; Sastry & Oore, 2020). These methods introduce extra computation to handle the OOD task. A previous study used the predicted maximum softmax probability (MSP) of the trained model directly to distinguish out-distribution data (Hendrycks & Gimpel, 2017). It is not only very simple but also achieves good results on some simple tasks. However, it does not perform as well as the advanced algorithms mentioned above on complex tasks. Inspired by the MSP algorithm, we extend their idea by using test time augmentation (TTA).

When testing deep neural networks, TTA utilizes the property that a well-trained classifier has a low variance in predictions across augmentations on most data (Shorten & Khoshgoftaar, 2019; Shanmugam et al., 2020). This phenomenon is only explored and verified on data drawn from the training distribution. However, this property does not generalize to other different distributions.

To verify, we use t-SNE (Van der Maaten & Hinton, 2008) to visualize the projected feature space of a ResNet-34 (He et al., 2016) supervisedly trained with CIFAR-10 (Krizhevsky et al., 2009) in Figure 1. The figure contains test samples from CIFAR-10 and SVHN Netzer et al. (2011) test sets. For each sample, we plot the raw feature and the feature of its augmented version with a line connecting them. It shows

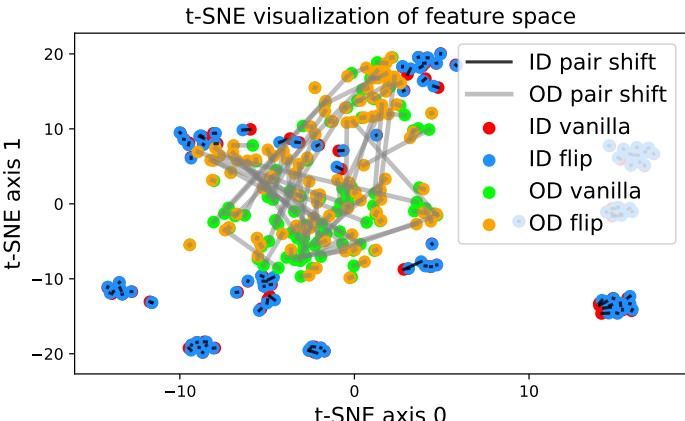

Figure 1: T-SNE feature space visualization. We randomly pick 200 CIFAR-10 (in-distribution, ID) and SVHN samples (out-distribution, OD). For each sample, we compute its vanilla feature and the feature of its augmented version. Here we use flip. We draw lines between a pair of the vanilla feature point and the augmented feature point. For each pair, the longer the line segment, the greater the feature difference.

that the relative distance between a CIFAR-10 pair (red-blue pairs with black lines) is statistically smaller than that of an SVHN pair (green-orange pairs with gray lines). Similar observations widely exist across networks, datasets, and augmentation methods (see Appendix A). Therefore, a well-trained model is *not* robust towards augmentations of all distributions; Instead, the property only holds for in-distribution data.

Based on this observation, we develop our algorithm, named TTACE (Test Time Augmentation and Consistency Evaluation), which has the following characteristics:

**Effectiveness.** TTACE achieves comparable or better detection performance on challenging benchmark vision datasets for both supervised and unsupervised SOTA models. **Efficiency.** TTACE is a post-hoc algorithm that is able to be directly applied to a plain classifier. It only computes two forward passes to obtain the anomaly scores, making it more efficient than most popular classifier-based algorithms. We provide a clock running time comparison in Figure 2. **Prior-free.** TTACE is dataset-agnostic and prior-free. Therefore, we do not have to tune the hyper-parameters with out-distribution prior, which may be hard or impossible to obtain during training. **Adaptability.** TTACE not only helps a plain classification model to detect anomalies but also adapts to other representation learning out-of-distribution detection algorithms (e.g., Tack et al.). Meanwhile, TTACE does not require access to internal features and gradients, making it desirable when a white-box model is unavailable.

We further analyze the reasons for improved performance through empirical observations. TTACE gains benefits from mutual relations between different augmentations of a single sample. We name all the classes except for the maximum class as *remaining classes*. Specifically, we verify that the mutual relations between the remaining classes of a single sample's different augmentations significantly diverge for in- and out-distribution data. The idea is different from previous ones which focus on maximum softmax probability (Hendrycks & Gimpel, 2017; Liang et al., 2018; Hendrycks et al., 2019), mutual relations between layers of a sample (Sastry & Oore, 2020), mutual relations between different samples (Tack et al., 2020) and class-conditional distributions (Lee et al., 2018; Winkens et al., 2020). We will provide detailed evidence for the huge statistical difference between the remaining classes on in-distribution and out-distribution data later.

To conclude, our contributions are:

- We observe a significant difference in the mutual relations between in- and out-distribution data, which is helpful for out-of-distribution detection.

- Based on this observation, we propose a new out-of-distribution detection algorithm, TTACE. It achieves comparable or better results on the challenging benchmark vision datasets for both su-

pervised and unsupervised dataset-vs-dataset anomaly detection. Meanwhile, it is computationally efficient and free to out-distribution prior and the model's internal weights/gradients.

- Empirical evidence for how TTACE works is provided, which is not often addressed by previous OOD algorithms.

## 2 Related work

**Out-of-distribution detection** The terminology is also named dataset-vs-dataset anomaly detection. A series of previous works focus on the dataset-vs-dataset image classification setting. Hendrycks & Gimpel observe that correctly classified samples tend to have larger maximum softmax probability, while erroneously classified and out-distribution samples have lower values. The distribution gap allows for the detecting anomalies. To enlarge the distribution gap, Hendrycks et al. enhance the data representations by introducing self-supervised learning and Liang et al. propose input pre-processing and temperature scaling. Input pre-processing advocates fine-tuning the input so that the model is more confident in its prediction. Temperature scaling introduces temperature to the softmax function. Besides maximum probability, Lee et al. introduce the Mahalanobis score, which computes the class-conditional Gaussian distributions from a pre-trained model and confidence scores. Tack et al. propose contrastive losses and a scoring function to improve out-of-distribution detection performance on high-resolution images. Sastry & Oore utilize Gram matrices of a single sample's internal features from different layers.

Each of these classifier-based dataset level algorithms has their own characteristics. Prior out-distribution knowledge is required to tune the parameters by some algorithms (Liang et al., 2018; Lee et al., 2018). Input pre-processing (Hsu et al., 2020; Lee et al., 2018; Liang et al., 2018) is time-consuming because one needs several times of forwarding and backpropagation. Scoring-based methods are computationally costly since they rely on internal model features (Abdelzad et al., 2019; Lee et al., 2018; Sastry & Oore, 2020). Our algorithm acts in a post-hoc manner, and utilizes test time augmentation and consistency evaluation to achieve prior-free and computational friendly out-of-distribution detection (summarized in Table 4).

**Data augmentations in out-of-distribution detection** Using data augmentations in out-of-distribution detection is not a piece of news. It has been incorporated by existing algorithms in both training and evaluating phases. Training data augmentation usually aims to learn a better data representation. Previous works employ augmentations from self-supervised learning and contrastive learning in training phase to gain better detection performance (Hendrycks et al., 2019; Tack et al., 2020; Zhou et al., 2021). Test data augmentation usually relates to the calculation of anomaly scores. Golan & El-Yaniv utilize summarized predicted probability over multiple geometric augmentations at test time to detect anomalies. Wang et al. calculate averaged negative entropy over each augmentation as the detection criterion.

Differently, our TTACE emphasizes the importance of interrelationships. It uses test data augmentation as well but with two main differences. First, we utilize the mutual relations between augmentations of a single sample rather than treat each augmentation independently, which is not explored by previous works. Second, we empirically verify that the performance improvement is attributed to the remaining classes, while previous works empirically find augmentations are helpful without any reasons.

Beyond the algorithms we mentioned above, various methods of out-of-distribution detection are well summarized by Chalapathy & Chawla. There are shallow methods such as OC-SVM (Schölkopf et al., 1999), SVDD (Tax & Duin, 2004), Isolation Forest (Liu et al., 2008), and deep methods such as deep SVDD (Ruff et al., 2018) and deep-SAD (Ruff et al., 2020) used to detect anomalies under different settings. We also notice that model uncertainty is highly related to out-of-distribution detection. In fact, out-of-distribution detection is exactly a task that assigns uncertainty (or anomaly) scores to samples, so the final goals are similar. The main factor that distinguishes them is how such scores are generated (Hendrycks et al., 2019; Seeböck et al., 2020).

---

**Algorithm 1** Computation pipeline of TTACE.

---

**Input:** A model $f$ pre-trained on in-distribution data; A predetermined data augmentation method $T$ drawn from function space $\mathcal{T}$; A set of test data $x_1, x_2, ..., x_n$ containing in-distribution and out-distribution data;

**Output:** Anomaly scores $S(x_1), S(x_2), ...S(x_n)$;

**for** $i = 1$ **to** $n$ **do**

    Augment $x_i$ with $T$ to obtain $T(x_i)$;

    Feed $x_i$ and $T(x_i)$ into $f$, and compute $f(x_i)$ and $f(T(x_i))$ through forward passes;

    $S(x_i) = 1 - \langle f(x_i), f(T(x_i)) \rangle$;

**end for**

---

## 3 Methodology

We present our simple yet effective algorithm TTACE, which is purely based on test time data augmentation and consistency evaluation. The whole computation pipeline is shown in Algorithm 1. Notice that there is no need to train a specific out-of-distribution detection model in our pipeline. Instead, the model in Algorithm 1 refers to a simple image classifier.

Our algorithm adapts to both supervised and unsupervised pre-trained models. The performance in both cases is covered in experiments. Below, we first explain the two cores of our algorithm in detail.

### 3.1 Data augmentation and feature space distance

We use data augmentation techniques without backpropagation at test time, which is similar to TTA, to help enlarge the sensitivity gap between in-distribution and out-distribution samples. Formally, given test data $X = \{\boldsymbol{x}_1, \boldsymbol{x}_2, \boldsymbol{x}_3, ..., \boldsymbol{x}_n\}$, a transformation function $T$ drawn from transformation function space $\mathcal{T}$, the augmentation process can be described as $\boldsymbol{x}_i \rightarrow T(\boldsymbol{x}_i)$.

Notice that unlike previous algorithms which also incorporate transformations, our algorithm focuses on the *relationship* between augmented feature pairs instead of treating each augmented output separately (Golan & El-Yaniv, 2018; Tack et al., 2020). So the transformations discussed below are actually pairs (the original sample and the transformed sample) instead of a single sample.

**Transformation function space** An immediate and important question is how to choose a suitable function space $\mathcal{T}$. We know that there are numerous different transformation functions designed for computer vision tasks. Testing all the transformations is exhausting. Fortunately, recent progress has revealed that modern neural networks naturally show sensitivity to low-frequency components (Wang et al., 2020) and visual chirality (Lin et al., 2020), especially for image tasks. Specifically, Wang et al. show that a well-trained convolution neural network can exploit high-frequency components of a figure that are imperceivable to human, and Lin et al. first challenge the flip-invariant assumption. Meanwhile, recall that the core idea of TTACE is to utilize the different degrees of sensitivity property of neural networks towards in- and out-distribution samples. Based on this intuition, our function space $\mathcal{T}$ contains Fast Fourier Transformation (FFT) and Horizontal Flip (HF). And TTACE extends these conclusions by revealing that while the neural networks are known to be sensitive to high frequency components and visual chirality (horizontal flip), the sensitivity is enhanced in out-distribution data.

FFT is a widely used efficient image processing technique (Bracewell & Bracewell, 1986; Oppenheim et al., 1997). FFT converts an image from the space domain to the frequency domain using a linear operator. Then the frequency representation can be inversely converted into the original image data by IFFT (Inverse Fast Fourier Transformation). In TTACE, we propose to cut off partial sensitive high-frequency signals. Our ablation experiments on the filter radius show that the out-of-distribution detection performance fluctuates little within a wide range of filter radius (see experiment section). In the following text, we will use $\text{FFT}_{100}$ to represent the FFT and IFFT transformation with a 100-pixel filter radius. Since Lin et al. discovered visual chirality, there have not been many studies exploring how to exploit it. We introduce it into out-of-distribution detection, and according to our observations, it indeed enlarges the sensitivity gap. We verify

| Algorithm | Out-distribution dataset | | | | | | AVG. | STD. |
|---|---|---|---|---|---|---|---|---|
| | CUB-200 | Dogs | Pets | Places | Caltech | DTD | | |
| Rot+Trans | $74.5_{\pm 0.5}$ | $77.8_{\pm 1.1}$ | $70.0_{\pm 0.8}$ | $53.1_{\pm 1.7}$ | $70.0_{\pm 0.2}$ | $89.4_{\pm 0.6}$ | 72.5 | 10.9 |
| CSI Unlabeled | $\mathbf{90.5}_{\pm \mathbf{0.1}}$ | $\mathbf{97.1}_{\pm \mathbf{0.1}}$ | $85.2_{\pm 0.2}$ | $78.3_{\pm 0.3}$ | $87.1_{\pm 0.1}$ | $\mathbf{96.9}_{\pm \mathbf{0.1}}$ | 89.2 | 6.6 |
| **TTACE**$_{\text{FFT100}}$ (ours) | $84.8_{\pm 0.1}$ | $93.9_{\pm 0.1}$ | $\mathbf{93.7}_{\pm \mathbf{0.1}}$ | $89.9_{\pm 0.2}$ | $91.1_{\pm 0.1}$ | $91.3_{\pm 0.1}$ | 90.8 | 3.0 |
| **TTACE**$_{\text{Flip}}$ (ours) | $86.3_{\pm 0.1}$ | $94.8_{\pm 0.1}$ | $94.7_{\pm 0.1}$ | $\mathbf{91.2}_{\pm \mathbf{0.2}}$ | $\mathbf{92.1}_{\pm \mathbf{0.1}}$ | $92.2_{\pm 0.2}$ | **91.9** | **2.8** |

Table 1: AUROC (%) of *ImageNet-30* results on unsupervised models. Higher average AUROC score is reached by TTACE. On Places-365, our algorithm improves the SOTA algorithm by 12.9%. We also report the average and standard variance of the AUROC values of all out-distribution datasets for each algorithm. Bold denotes the best results.

through extensive experiments (Figure 1 and Appendix A) to show that the sensitivity is more significant in out-distribution data with both transformations. Specifically, the distance of paired in-distribution sample features is statistically shorter than that of paired out-distribution samples.

**Remark** All the discussions about the transformation functions above focus on the testing phase rather than the training phase. There is no apparent relationship between the transformation functions used when training the network and our transformation function space $\mathcal{T}$ at test time as far as we know. Exploring such relationship is an interesting future work.

Other transformations are also worth a try. There are also other transformations which may satisfy the sensitivity criterion such as adversarial examples. However, they may bring undesired randomness and high computation burden, which is contrary to the original intention of TTACE. Additionally, employing data augmentation is not a new idea in out-of-distribution detection as mentioned in related works. Curiously, we implement rotation transformation in Appendix C, which is proposed by a previous work (Hendrycks et al., 2019), although no previous researches reported networks' sensitivity of rotation as far as we know.

### 3.2 Consistency Evaluation

Unlike previous works that ignore the interaction between different transformations of a sample, our anomaly score evaluates the relations between augmentations of a single sample in a simple but effective form. With a model $f$ trained on the training (in-distribution) data, we denote the softmax output of $\boldsymbol{x}$ as $f(\boldsymbol{x})$. Suppose we fix a transformation method $T$. The consistency score of a given data $\boldsymbol{x}$ is defined as the inner product of the model output of $\boldsymbol{x}$ and $T(\boldsymbol{x})$, which is $\langle f(\boldsymbol{x}), f(T(\boldsymbol{x})) \rangle$.

To further enlarge the gap of the model output for in- and out-distribution data, we use a fixed temperature scaling technique. Given a sample $\boldsymbol{x}$, we denote the softmax output with temperature scaling parameter $t$ as $f(\boldsymbol{x}; t)$. And, define our final anomaly score for a given sample $\boldsymbol{x}$ as

$$S(\boldsymbol{x}) = 1 - \langle f(\boldsymbol{x}; t), f(T(\boldsymbol{x}; t)) \rangle. \tag{1}$$

Based on the observation (Figure 1), the model $f$ tends to make more consistent outputs for in-distribution data than out-distribution data. Ideally, $S(\boldsymbol{x})$ should be close to 0 for in-distribution data and $S(\boldsymbol{x})$ should be approximately 1 for out-distribution data.

## 4 Experiments

### 4.1 Baseline algorithms and metrics

In out-of-distribution detection, algorithms with different settings often cannot be measured simultaneously. For example, some algorithms focus on the one-class-vs-all setting (Ruff et al., 2018; Golan & El-Yaniv, 2018; Wang et al., 2019; Gong et al., 2019) (a.k.a. anomaly detection), while others focus on the dataset-vs-dataset setting (Liang et al., 2018; Zhou et al., 2021) only. There are also algorithms that cover multiple

settings (Hendrycks et al., 2019; Tack et al., 2020). Since our algorithm relies on the remaining classes, i.e., multiple classes in the training set, those one-vs-all algorithms are not compared. The SOTA algorithm of advanced dataset is CSI (Tack et al., 2020). It mainly benefits from better representation learned by contrastive learning. We also cover MSP (Hendrycks & Gimpel, 2017), ODIN (Liang et al., 2018), Mahalanobis (Lee et al., 2018), and Rot (Hendrycks et al., 2019) as baselines. The Rot algorithm also uses the transform during the test phase but in a different manner. Specifically, it uses the negative log probability assigned to the true rotation to improve performance.

Following previous works (Hendrycks et al., 2019; Lee et al., 2018; Tack et al., 2020), we use the AUROC (Area Under Receiver Operating Characteristic curve) as our evaluation metric. It summarizes True Positive Rate (TPR) against False Positive Rate (FPR) at various threshold values. The use of the AUROC metric frees us from fixing a threshold and provides an overall comparison metric.

### 4.2 Dataset settings

Dataset settings are diverse in out-of-distribution detection. We mainly consider the advanced ImageNet (Deng et al., 2009) settings. ImageNet-30, a subset of ImageNet, is the commonly used benchmark ImageNet setting. Beyond that, we also conduct experiments on other larger ImageNet settings, including the full set settings and three subset settings, to evaluate the generalization ability of TTACE.

The *ImageNet-30* subset benchmark is used in relevant researches (Hendrycks et al., 2019; Tack et al., 2020). The corresponding out-distribution datasets includes CUB-200 (Wah et al., 2011), Stanford Dogs (Khosla et al., 2011), Oxford Pets (Parkhi et al., 2012), Places-365 (Zhou et al., 2018) with small images (256*256) validation set, Caltech-256 (Griffin et al., 2007), and Describable Textures Dataset (DTD) (Cimpoi et al., 2014).

For the full-size ImageNet setting, the whole validation set is treated as the in-distribution test set. Since the ImageNet training set already contains all-embracing natural scenery, the corresponding out-distribution dataset is chosen to be a publicly available artificial image dataset from Kaggle[1]. We use the first 50000 figures since the ImageNet validation dataset is of the same size. Some examples of the dataset are depicted in Appendix B. We will call this setting *ImageNet vs. Artificial*. We also conduct experiments on variants of ImageNet such as ImageNet-R and ImageNet-A in Appendix C.

Beyond that, we also validate our algorithm on several public ImageNet subset combinations introduced by other works (Geirhos et al., 2019; Tsipras et al., 2020) and public sources[2]. These additional three ImageNet subsets are denoted as *Living 9*, *Geirhos 16*, and *Mixed 10*. The out-distribution categories are randomly selected from the complement set. The detailed subset hierarchy and configuration are shown in Appendix B.

We notice the performance of out-of-distribution detection is affected by the trained classifiers. So we train networks with three random seeds per setting and report the mean and variance. The training details are in Appendix G.

### 4.3 Detection results

This section covers the main out-of-distribution detection results including the ImageNet-30 (Table 1), the full ImageNet setting (Table 2), and three ImageNet subset settings(Table 3).

**ImageNet-30** For the *ImageNet-30* benchmark, we adapt the TTACE to the unsupervised CSI model by substituting the predicted probabilities in Eqn. (1) with output features (512 dimensional tensors). The results are shown in Table 1. TTACE outperforms SOTA algorithms in three of the OOD datasets and reaches a better average AUROC value. At the same time, TTACE achieves the smallest average standard variance, indicating that it exhibits robustness to out-distribution dataset shift.

---

[1] https://www.kaggle.com/alamson/safebooru
[2] https://github.com/MadryLab/robustness

| Algorithm | Architecture | |
|---|---|---|
| | ResNet-50 | DenseNet-121 |
| MSP | 81.05 | 79.81 |
| **TTACE**$_{\text{FFT100}}$ (ours) | **92.08** | **90.84** |
| **TTACE**$_{\text{Flip}}$ (ours) | 90.20 | 89.36 |

Table 2: AUROC (%) of full *ImageNet vs. Artificial* dataset on the pre-trained torchvision models (Paszke et al., 2019).

| Architecture | Algorithm | ImageNet subset settings | | |
|---|---|---|---|---|
| | | Living 9 | Geirhos 16 | Mixed 10 |
| ResNet-50 | MSP | $95.32_{\pm 0.15}$ | $90.06_{\pm 1.37}$ | $95.29_{\pm 0.06}$ |
| | ODIN | $96.78_{\pm 0.36}$ | $82.35_{\pm 0.70}$ | $95.92_{\pm 0.78}$ |
| | Mahalanobis[1] | $64.21_{\pm 3.31}$ | $64.31_{\pm 1.96}$ | $71.33_{\pm 8.11}$ |
| | **TTACE**$_{\text{FFT100}}$ (ours) | $97.17_{\pm 0.20}$ | $92.53_{\pm 0.67}$ | $97.25_{\pm 0.13}$ |
| | **TTACE**$_{\text{Flip}}$ (ours) | $\mathbf{97.29_{\pm 0.19}}$ | $\mathbf{92.82_{\pm 0.68}}$ | $\mathbf{97.38_{\pm 0.15}}$ |
| DenseNet-121 | MSP | $95.57_{\pm 0.07}$ | $91.76_{\pm 0.09}$ | $94.99_{\pm 0.40}$ |
| | ODIN | $96.61_{\pm 0.52}$ | $86.61_{\pm 4.36}$ | $96.49_{\pm 0.83}$ |
| | Mahalanobis[2] | $49.01_{\pm 2.95}$ | $59.26_{\pm 5.83}$ | $65.52_{\pm 3.88}$ |
| | **TTACE**$_{\text{FFT100}}$ (ours) | $97.21_{\pm 0.07}$ | $93.10_{\pm 0.24}$ | $\mathbf{96.82_{\pm 0.35}}$ |
| | **TTACE**$_{\text{Flip}}$ (ours) | $\mathbf{97.26_{\pm 0.08}}$ | $\mathbf{93.23_{\pm 0.22}}$ | $96.80_{\pm 0.33}$ |

[1,2] We use the official code and hyper parameter settings of the paper. The implementation details are in Appendix F.

Table 3: AUROC (%) of ImageNet subset results on supervised models. TTACE has a uniform performance improvement on all settings. Bold denotes the best results.

**ImageNet**  For the full *ImageNet vs. Artificial* setting, we directly download the pre-trained model from Pytorch. Since the weights of the downloaded model are trained for image classification, for fair comparision, we only compare our post-hoc TTACE with the MSP algorithm. The performance is shown in Table 2.

**ImageNet subsets**  For the other three subset settings, *Living 9*, *Geirhos 16*, and *Mixed 10*, we compare TTACE with supervised algorithms listed in Table 3. We can see that TTACE is more efficient and outperforms all counterparts. All the compared algorithms are implemented according to the original paper and code except necessary modification of ImageNet figure size. Implementation details are stated in Appendix F.

**Discussion**  Zhou et al. Zhou et al. (2021) report that severe performance degradation of the Mahalanobis (Lee et al., 2018) exists when generalizing to other datasets due to validation bias. In order to verify whether TTACE also has similar problems, we conduct experiments on CIFAR-10 based settings to test the generalization ability of TTACE. Limited to space, the detection results are in Appendix H. Our experiments lead to a similar conclusion as Zhou et al. that simpler algorithms generalize better on different settings. The results also show that TTACE can not only achieve better results on advanced datasets, but also has stable generalization performance on low-resolution datasets such as CIFAR-10.

## 4.4  Algorithms running time

We show the computational costs of all considered algorithms in this section since running time is important in practical use. We run all experiments on an RTX3090 graphics card.

The post-hoc algorithms, ODIN and TTACE, do not include a training phase, while the Mahalanobis needs a little bit of time to train. All implementations are based on official code sources. Figure 2 shows that TTACE only requires $10\% \sim 40\%$ running time of advanced algorithms while maintaining comparable or better

| Algorithm | Access to the internal weights | Out-distribution data prior | Parameters search | Computational cost |
|---|---|---|---|---|
| ODIN (Liang et al., 2018) | **Not Req.** | Req. | Req. | 1FP+1BP+1FP |
| Mahalanobis (Lee et al., 2018) | Req. | Req. | Req. | $1FP+k\times(1FP+1BP+1FP)^1+Regression^2$ |
| **TTACE** (ours) | **Not Req.** | **Not Req.** | **Not Req.** | **2FP** |

[1] Compute feature mean and covariance of k selected layers.
[2] Regression on the computed Mahalanobis score.

Table 4: Training and testing costs of highly related (classifier-based) algorithms. Running time experiments and detailed explanations are shown in Figure 2 and discussed in the algorithm running time section. TTACE actis in a post-hoc manner and only requires two forward passes to get the score. (FP: forward pass. BP: backpropagation. Req.: Required.)

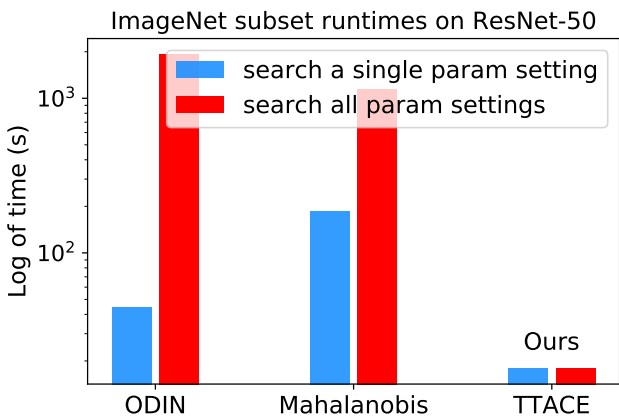

Figure 2: Average classifier-based algorithm running time on ImageNet subset *Living 9*. For algorithms that require parameter search, we provide the total time of searching all parameter settings and the time of a single search. TTACE has the minimum time consumption.

detection results (Table 3). The difference in running time is mainly due to the use of different techniques as summarized in Section 2 and Table 4. The number of parameter searching groups comes directly from the original papers and codes. The running time of CSI is missing because introducing contrastive learning is often considered much more time-inefficient than plain training which is the default training methods of the three algorithms in Figure 2. The details, criteria for choosing comparison algorithms and running times on other network architectures are stated in Appendix E.

### 4.5 Ablation studies for filter radius and temperature

Performance tables in this paper include several different augmentation methods of TTACE. Although the AUROC values fluctuate with different augmentation methods and filter radius, the performance is similar. With linear searching across the FFT filter radius and temperature settings, we demonstrate that they have a small effect on the out-of-distribution detection performance within a wide range.

For the FFT filter radius, we provide ablation studies across a wide range of filter radius settings, ranging from 40 to 160 pixels, in Figure 3. Among all settings, TTACE does not degrade much and still outperforms other algorithms as shown in Table 3, which demonstrates that the performance of TTACE is not very sensitive to the filter radius. A possible reason is that the energy is highly concentrated in the low frequency space instead of high frequency parts for images. More ablations of other settings are in Appendix I.

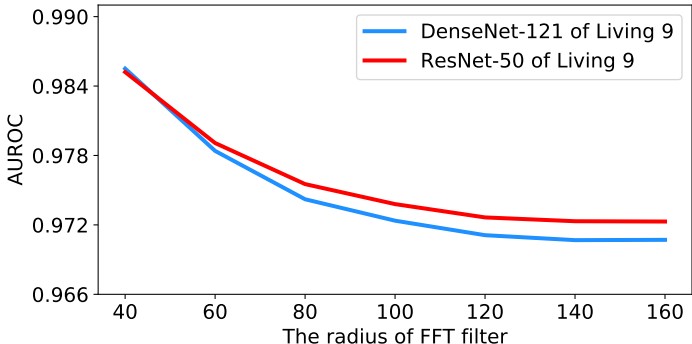

Figure 3: Detection AUROC with different radius of the FFT filter. Flatter curves mean less sensitivity to parameters.

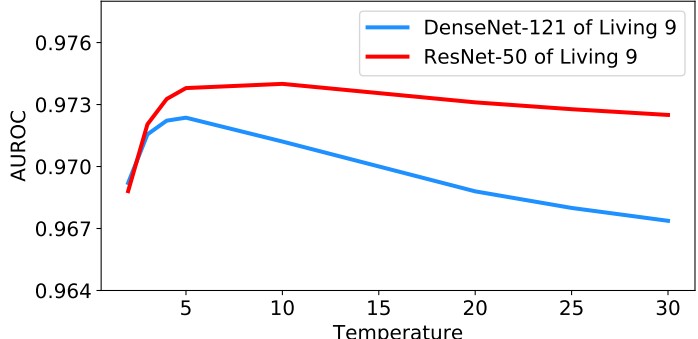

Figure 4: Detection AUROC with different temperature. Flatter curves mean less sensitivity to parameters.

Unlike the previous algorithms (Lee et al., 2018; Liang et al., 2018), which consider large-scale candidates of temperature scaling (e.g., from 1 to 1000) and then do the grid-search with prior out-distribution knowledge, we empirically find that the best temperature for our method is stable around 5 and the performance does not decrease much within the considered wide temperature range. All our experiments set the temperature parameter to 5 if not specified. Ablation studies for temperature parameter are shown in Figure 4, and more results on other settings can be found in Appendix J. All curves show that TTACE is not sensitive to the temperature parameter, thus we do not have to tune it with out-distribution prior.

## 5 Empirical analysis

In this section, we will explain where performance improvement comes from empirically.

### 5.1 Empirical analysis of using remaining classes probability

Our anomaly score (Eqn (1)) adds up all class probabilities rather than focusing only on the maximum predicted probability (Hendrycks & Gimpel, 2017; Hendrycks et al., 2019). The main difference between consistency evaluation and maximum predicted probability comes from the remaining score $S_{rem}$, defined as

$$S_{rem} = \langle f(\boldsymbol{x}), f(T(\boldsymbol{x})) \rangle - f_j(\boldsymbol{x})f_j(T(\boldsymbol{x})), \tag{2}$$

where $j = \arg\max f_j(\boldsymbol{x})$ is the predicted class with the highest probability of the input $\boldsymbol{x}$.

Figure 5 shows the beneficial effect of including the remaining classes. First, Figure 5(a) presents the distribution of maximum probability $f_j(\boldsymbol{x})$. Notice that the x axis is evenly divided into several intervals. Then, for samples within each interval, we calculate the mean and variance of the remaining scores $S_{rem}$, as

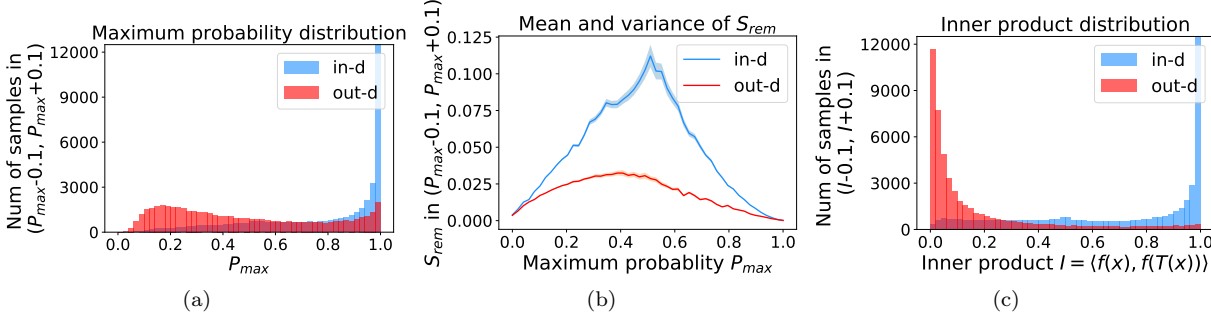

Figure 5: Figures showing the effect of remaining classes of a pre-trained ResNet-50. Full ImageNet valida-tion set and the Artificial dataset (see Section 4.2) are treated as in- and out-distributions respectively. (a) Maximum probability distributions of in- and out-distribution samples. (b) Mean and variance of remaining scores within each slot $(P_{max} - \epsilon, P_{max} + \epsilon)$. (c) Distributions of inner product $I$ of in- and out-distribution samples. Note that our anomaly score $S(x) = 1 - I$.

Figure 5(b) shows, where the line is the average and the shaded area is variance. More specifically, to draw the Figure 5(b), we first evenly divide the interval $(0, 1)$ in Figure 5(a) into 50 parts. Then we pick one interval $(P_{max} - \epsilon, P_{max} + \epsilon)$, select all the samples whose maximum predicted probability $f_j(\boldsymbol{x})$ is between $(P_{max} - \epsilon, P_{max} + \epsilon)$, and compute the mean and variance of those selected samples' remaining scores $S_{rem}$. Notice that the reaming classes does not contain class $j$. For every interval, we process the in- and out-distribution data separately indicated by the red and blue lines. Recall that the goal of out-of-distribution detection is

to decrease the overlap sample number from the two distributions. Intuitively, samples in Figure 5(a) from both in- and out-distributions with similar $P_{max}$ values (i.e., samples from the same interval) are mixed, which hinders out-of-distribution detection. Figure 5(b) shows that for samples within the same interval, the remaining scores of the in-distribution samples are statistically larger than that of out-distribution samples. Therefore, adding the remaining scores gives different momentum to the mixed samples located in the same interval in Figure 5(a), and the momentum of the in-distribution samples is larger (Figure 5(b)), which makes the two distributions that originally coincided to be distinguished. When this process happens in every interval, we obtain our inner product scores $(1 - S(x))$ as depicted in Figure 5(c), which improves the maximum probability method from Figure 5(a). Comparing the maximum probability (Figure 5(a)) and consistency evaluation methods (Figure 5(c)), our method is more discriminative. Generally, a larger vertical gap in Figure 5(b) indicates a higher performance improvement. This empirical analysis is consistent with our numerical results in the experiment section. Similar observations across network architectures and datasets can be found in Appendix D.

## 6 Limitation

Although TTACE is effective and easy to be implemented, it has its inherit limitations. TTACE may fail under some adversarially designed synthetic datasets, e.g., a transformation-invariant vision dataset. For example, it is obvious that the HF transformation takes no effect when we artificially design a mirror symmetrical dataset. In this case, we have $f(\boldsymbol{x}) = f(T(\boldsymbol{x}))$ and the feature space distance like Figure 1 will not be sensitive to the in- and out-distribution samples and the performance will decrease. Fortunately, we have not met such adversarial cases in real-world datasets.

## 7 Conclusion

In this paper, we propose an elegant (simple yet effective) dataset-vs-dataset vision post-hoc out-of-distribution detection algorithm, TTACE, which reaches comparable or better detection performance with current advanced and SOTA algorithms. It largely reduces the computational costs, demands no specific

network architectures, no access to the internal feature, and no prior out-distribution data knowledge. We empirically show that TTACE gains performance improvement from the remaining classes, which are ignored by some previous works.

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

## A  Test time augmentation feature space under different data distributions

We verify the observation mentioned in Section 1 across architectures, dataset settings, and augmentation methods in Figure 6. Due to the resolution problem, some images may be needed to zoom in to reflect the fine details we described. For each figure, we randomly pick 100 in-distribution samples and 100 out-distribution samples. For each pair of samples, a longer line segment means the distance in the projected feature space is large, leading to a smaller inner product value. In general, all the out-distribution samples in each subfigure have a statistically larger feature difference, compared to in-distribution samples. The dataset names, architectures, and transformation methods are annotated in each title of the subfigures.

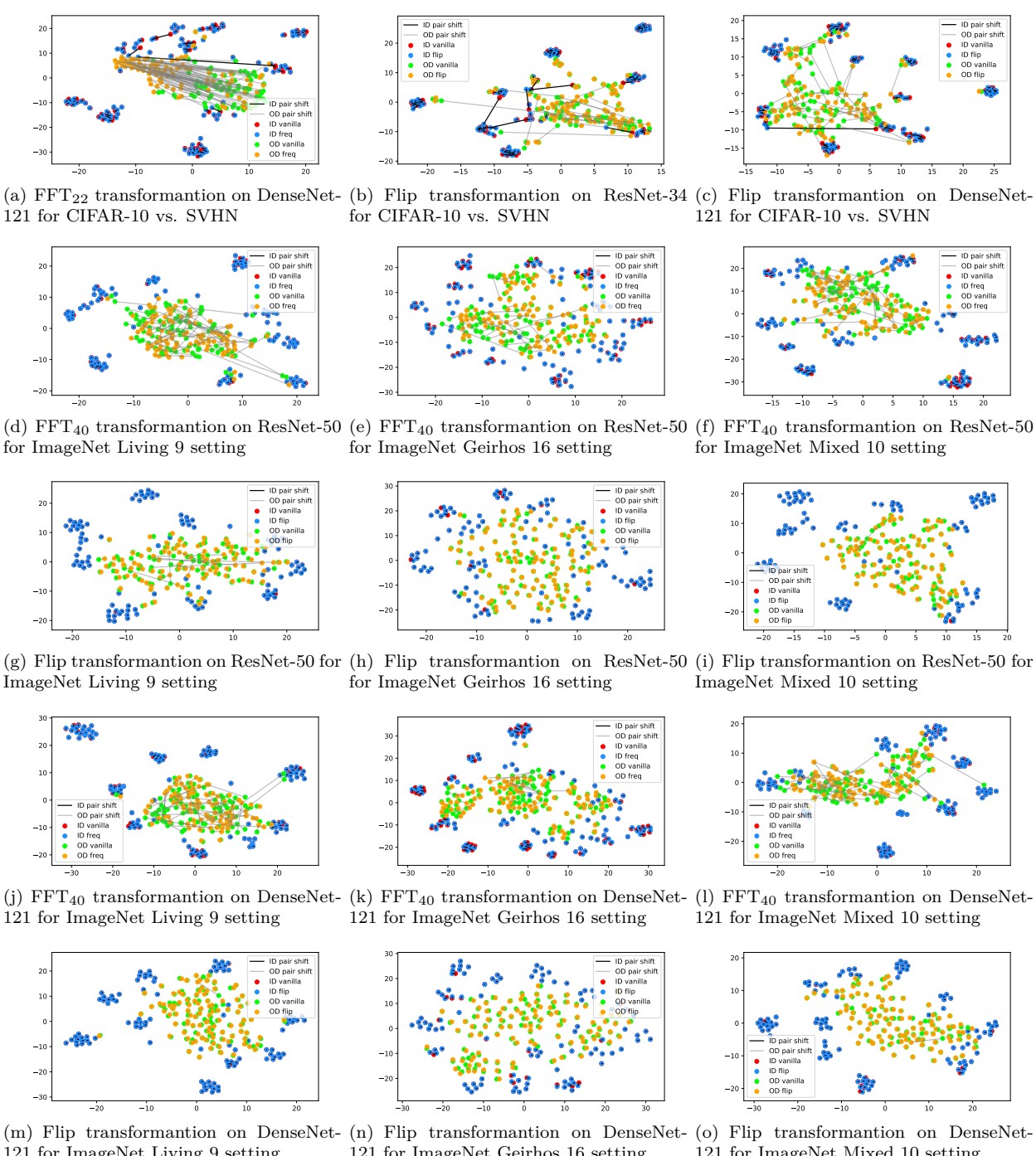

(a) FFT$_{22}$ transformantion on DenseNet-121 for CIFAR-10 vs. SVHN

(b) Flip transformantion on ResNet-34 for CIFAR-10 vs. SVHN

(c) Flip transformantion on DenseNet-121 for CIFAR-10 vs. SVHN

(d) FFT$_{40}$ transformantion on ResNet-50 for ImageNet Living 9 setting

(e) FFT$_{40}$ transformantion on ResNet-50 for ImageNet Geirhos 16 setting

(f) FFT$_{40}$ transformantion on ResNet-50 for ImageNet Mixed 10 setting

(g) Flip transformantion on ResNet-50 for ImageNet Living 9 setting

(h) Flip transformantion on ResNet-50 for ImageNet Geirhos 16 setting

(i) Flip transformantion on ResNet-50 for ImageNet Mixed 10 setting

(j) FFT$_{40}$ transformantion on DenseNet-121 for ImageNet Living 9 setting

(k) FFT$_{40}$ transformantion on DenseNet-121 for ImageNet Geirhos 16 setting

(l) FFT$_{40}$ transformantion on DenseNet-121 for ImageNet Mixed 10 setting

(m) Flip transformantion on DenseNet-121 for ImageNet Living 9 setting

(n) Flip transformantion on DenseNet-121 for ImageNet Geirhos 16 setting

(o) Flip transformantion on DenseNet-121 for ImageNet Mixed 10 setting

Figure 6: More feature space projection examples across networks, datasets and augmentation methods.

# B    Dataset construction and examples

We show the detailed WordNet IDs and corresponding classes of each ImageNet subset in Table 5. And some examples of the Artificial dataset are shown in Figure 7.

| Subsets | In-distribution classes & WordNet IDs | | Out-distribution classes & WordNet IDs | |
|---|---|---|---|---|
| | dog | n02084071 | furniture | n03405725 |
| | bird | n01503061 | oven | n03862676 |
| | arthropod | n01767661 | aircraft | n02686568 |
| | reptile | n01661091 | bicycle | n02834778 |
| Living 9 | primate | n02469914 | musical instrument | n03800933 |
| | fish | n02512053 | | |
| | feline | n02120997 | | |
| | bovid | n02401031 | | |
| | amphibian | n01627424 | | |
| | aircraft | n02686568 | insect | n02159955 |
| | bear | n02131653 | salamander | n01629276 |
| | bicycle | n02834778 | clothing | n03623556 |
| | bird | n01503061 | dophin | n02068974 |
| | boat | n02858304 | reptile | n01661091 |
| | bottle | n02876657 | | |
| | car | n02958343 | | |
| | cat | n02121808 | | |
| Geirhos 16 | char | n03001627 | | |
| | clock | n03046257 | | |
| | dog | n02084071 | | |
| | elephant | n02503517 | | |
| | keyboard | n03614532 | | |
| | knife | n03623556 | | |
| | oven | n03862676 | | |
| | truck | n04490091 | | |
| | dog | n02084071 | furniture | n03405725 |
| | bird | n01503061 | fish | n02512053 |
| | insect | n02159955 | knife | n03623556 |
| | monkey | n02484322 | keyboard | n03614532 |
| Mixed 10 | car | n02958343 | | |
| | feline | n02120997 | | |
| | truck | n04490091 | | |
| | fruit | n13134947 | | |
| | fungus | n12992868 | | |
| | boat | n02858304 | | |

Table 5: ImageNet subsets classes and the corresponding WordNet IDs.

## C  Additional experiments

We conduct additional experiments. Limited to space, we put these less relevant experiments here.

First of all, we evaluate *ImageNet vs. ImageNet-R* and *ImageNet vs. ImageNet-A* with ResNet-50 and DenseNet-121. The classifier parameters are downloaded from Pytorch official implementation. The AUROC values are shown in Table 6 and Table 7. Although TTACE achieves the best performance, we are not sure whether treating ImageNet-R and ImageNet-A as out-distribution datasets is appropriate since they share the same categories as ImageNet. In the authors' opinion, out-of-distribution detection is a task where the in- and out-distribution do not share the same categories. And the categories of in- and out-distribution public benchmark dataset and the dataset settings in our main text are different.

Secondly, we try to use rotation as the transformation function in TTACE (denoted as **TTACE**$_{\text{Rot*}}$). Notice that we **do not** propose to use rotation in practice. We perform this experiment because the rotation transformation has been reported to be effective with another technique in other works Hendrycks et al.

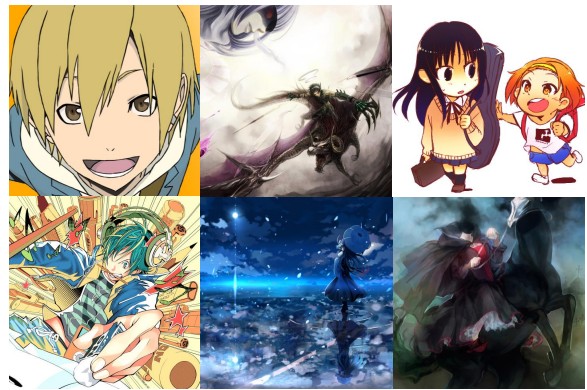

Figure 7: Some examples from the Anime dataset.

| Algorithms | ImageNet-R | ImageNet-A |
|---|---|---|
| MSP | 80.55 | 82.49 |
| **TTACE**$_{FFT100}$ (ours) | **87.97** | 84.68 |
| **TTACE**$_{Flip}$ (ours) | 87.74 | **85.21** |

Table 6: AUROC values on ResNet-50 of *ImageNet vs. ImageNet-R* and *ImageNet vs. ImageNet-A*. Bold denotes the best results.

(2019). However, since the core rule of the transformation space is sensitivity as discussed in Section 3.1, and we are not aware of any work that studies the sensitivity of neural networks to rotation, it may not be a suitable transformation. Nevertheless, we report the results in Table 7 to interested readers.

Thirdly, we also implement the original Rot+Trans algorithm on the full ImageNet and three subset settings. We do not put the results into the main text since the model parameter is separately trained apart from the model in Table 3. We directly apply the original Rot+Trans to the following settings. The results are shown in Table 9.

# D    Remaining score effect

More examples showing the effect of the remaining classes across network architectures and datasets are shown in Figure 8 and Figure 9. Similar to Figure 5, each row contains (a) Maximum probability distributions of in- and out-distribution samples. (b) Mean and variance of remaining scores within each slot ($P_{max} - \epsilon, P_{max} + \epsilon$). (c) Inner product (ours) distributions of in- and out-distribution samples.

# E    Detailed computational cost description of highly related algorithms

In Table 4, we briefly summarize the extra training and testing computational cost of highly related (standard classifier-based) algorithms. But we did not provide details because of space limitation. In this section, we will describe the details to clarify.

## E.1    ODIN

After obtaining a standard classification model, ODIN requires input pre-processing, which first needs a forward pass to compute the loss, then a backpropagation to modify the input, and finally a forward pass again to calculate the final predicted probability values. The searching process of hyperparameters like noisy magnitude and temperature demands prior out-distribution knowledge.

| Algorithms | ImageNet-R | ImageNet-A |
|---|---|---|
| MSP | 78.82 | 83.29 |
| **TTACE**$_{FFT100}$ (ours) | **86.27** | 84.76 |
| **TTACE**$_{Flip}$ (ours) | 85.71 | **85.33** |

Table 7: AUROC values on DenseNet-121 of *ImageNet vs. ImageNet-R* and *ImageNet vs. ImageNet-A*. Bold denotes the best results.

| Algorithms | ResNet-50 | DenseNet-121 |
|---|---|---|
| **TTACE**$_{Rot90}$ | 81.53 | 80.29 |
| **TTACE**$_{Rot180}$ | 82.71 | 80.32 |
| **TTACE**$_{Rot270}$ | 81.58 | 82.07 |

Table 8: AUROC values on ResNet-50 and DenseNet-121 of *ImageNet vs. Artificial*. Note that no previous work has reported the sensitivity of networks to rotation transformation, so we do not recommend incorporating rotation into the transformation function space.

### E.2  Mahalanobis

Mahalanobis method assumes that the features of the in-distribution dataset follow a Gaussian distribution. It first needs forward all training samples to compute feature mean and variance. And then, it adopts a similar idea from input pre-processing, thus requires two forward passes and one backward pass to record the internal feature at test time. Meanwhile, the Mahalanobis method does a grid search across k (predetermined) internal layers and several noise magnitude values. The best setting is selected with the help of prior out-distribution knowledge. After recording all the features of the test samples, it finally trains a regression model and gives final anomaly scores of test samples.

### E.3  Ours

The computational cost of TTACE is almost the same as doubling testing a test point. We only need two forward passes per sample to compute the anomaly score. The pipeline is shown in Algorithm 1.

### E.4  Running time of classifier based algorithms

For experiments in this section, we run all the running time experiments on an RTX 2080Ti graphic card. We only consider the CIFAR-10 vs. SVHN setting since the proportion of running time does not change with different datasets. For DenseNet-100, we conduct experiments similarly and the result is shown in Figure 10. And we omit the common classifier training time.

For ODIN, we slightly simplify the search space. According to the original ODIN paper, there are 210 times of searches in total. We only choose 44 among the 210 settings, which are useful in the CIFAR-10 vs. SVHN setting Liang et al. (2018). The Mahalanobis algorithm demands recording the mean and variance of the training samples, which takes about 19 seconds for the CIFAR-10 dataset in ResNet-34. This process will take more time if the training sample size increases. We follow the hyperparameter searching settings, which include 7 independent searches.

We do not mention other algorithms because they adopt other training ways which are significantly different from supervised training. For example, GAN-based algorithms require GAN training, which highly depends on the GAN algorithm. And obtaining a satisfying GAN usually takes a longer time than supervised classifier training. Contrastive learning aided algorithms Tack et al. (2020) use contrastive methods (SimCLR Chen et al. (2020)) to train a model, which takes much more time than standard classifier training. For example, SimCLR still gains performance improvement after 800 epochs training Chen et al. (2020) and the computation of contrastive losses is more complicated. Considering the significant difference in pre-training time, we

| Algorithms | *ImageNet vs Artificial* | *Living 9* | *Geirhos 16* | *Mixed 10* |
|---|---|---|---|---|
| Rot+Trans | 56.94 | 55.90 | 56.27 | 64.50 |

Table 9: AUROC values of Rot+Trans algorithm on *ImageNet vs. Artificial*, *Living 9*, *Geirhos 16*, and *Mixed 10*.

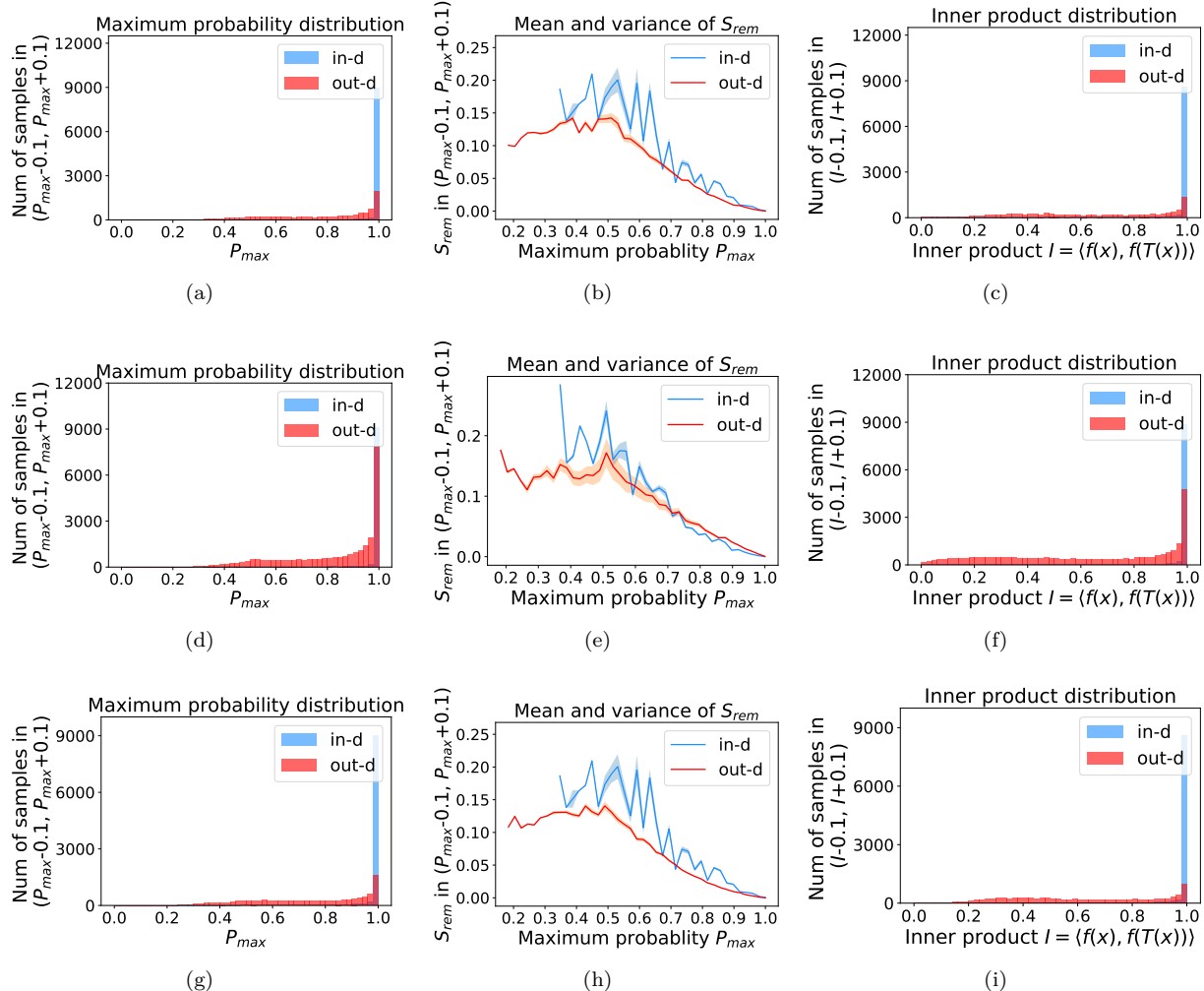

Figure 8: Figures showing the effect of remaining classes of a pre-trained model. (a)-(c): CIFAR-10 vs. ImageNet_resize on DenseNet-100. (d)-(f): CIFAR-10 vs. SVHN on ResNet-34. (g)-(i): CIFAR-10 vs. LSUN_resize on DenseNet-100.

only compare the running time of our algorithm with these relatively lightweight standard classifier-based algorithms.

# F   Attempts for adapting previous algorithm

## F.1   Network architecture

For CIFAR classifiers, we simply follow the common architectures which are also used by Liang et al. Liang et al. (2018) and Lee et al. Lee et al. (2018).

Figure 9: Figures showing the effect of remaining classes of a pre-trained model on ImageNet subsets. (a)-(c): The Living 9 subset setting on ResNet-50. (d)-(f): The Geirhos 16 subset setting on DenseNet-121. (g)-(i): The Mixed 10 subset setting on ResNet-50.

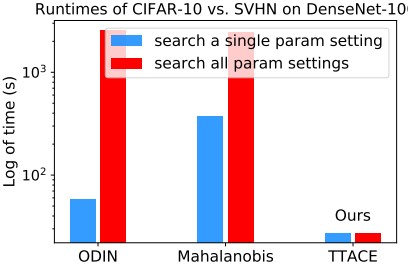

Figure 10: Average algorithm running time on DenseNet-100.

CIFAR models can not be directly used in ImageNet images. Following common configurations, we resize the ImageNet figures into 224x224, and use *torchvision*'s official ImageNet models: ResNet-50 and DenseNet-121 from https://pytorch.org/vision/stable/models.html. Standard normalization is added when pre-

processing the images. For ImageNet subset training, we only change the output size of the last layer to the corresponding class numbers.

## F.2 MSP

The MSP algorithm Hendrycks & Gimpel (2017) can be directly adopted to different datasets because its criteria is the maximum probability value. And stable performance across low-resolution datasets and high-resolution datasets can be observed in Table 2, Table 3 and Table 12, .

## F.3 ODIN

The major difference between ODIN Liang et al. (2018) and the MSP method is that ODIN uses two techniques, input-preprocessing and temperature scaling, to enlarge the gap between the in-distribution and out-distribution samples in maximum predicted value. For both CIFAR and ImageNet subset settings, we do a grid search for noise value from [0, 0.0005, 0.001, 0.0014, 0.002, 0.0024, 0.005, 0.01, 0.05, 0.1, 0.2], and temperature value from [1, 10, 100, 1000].

ODIN and the MSP method do not tune the feature space of deep neural networks, and reach good performance in both the ImageNet subset settings (Table 3) and CIFAR (Table 12).

## F.4 Mahalanobis algorithm

For the CIFAR setting, we directly use the official setting mentioned in Lee et al. (2018). For the ImageNet subset settings, we use the official ResNet-50 and DenseNet-121 architecture from torchvision. The choice of internal output features is the same as the CIFAR setting: we choose the output of the first convolution layer and output of every transition layer for DenseNet, and the input of the first residual block and the output of every residual block for ResNet.

We also expand the hyperparameter search space. Specifically, we search the noise magnitude of the Mahalanobis method from [0.0, 0.5, 0.2, 0.1, 0.05, 0.02, 0.01, 0.005, 0.002, 0.0014, 0.001, 0.0005], which expands the original search space a lot. The validation dataset size is defined as 10% of the whole test set size, which is the same as the original paper.

## G   Classifier model training

Since we find the performance of classifier-based algorithms is highly based on the pre-trained classifier, e.g. different random seeds, we train three models for all settings and report the mean and variance. Standard training data augmentation methods are adopted, including resizing, padding and flipping. For full ImageNet training, we directly use pre-trained models from torchvision[3]. The validation accuracy is 76.01% for ResNet-50 and 74.47% for DenseNet-121.

For ImageNet subset training, we use the official implementation of ResNet-50 and DenseNet-121 models from *torchvision*. We train 200 epochs using SGD with a stepping learning rate. The training and test accuracy is shown in Table 10.

| Subset | ResNet-50 | | DenseNet-121 | |
|---|---|---|---|---|
| | Training | Test | Training | Test |
| Living 9 | $96.00 \pm 0.35$ | $75.91 \pm 0.70$ | $96.23 \pm 0.05$ | $76.86 \pm 0.35$ |
| Geirhos 16 | $93.56 \pm 2.79$ | $64.61 \pm 0.10$ | $96.39 \pm 0.11$ | $67.06 \pm 0.44$ |
| Mixed 10 | $95.37 \pm 0.16$ | $80.37 \pm 0.13$ | $95.90 \pm 0.26$ | $80.24 \pm 0.46$ |

Table 10:   Classifier training and test accuracy (%) on ImageNet subset.

---

[3]https://pytorch.org/docs/stable/torchvision/models.html

For the CIFAR-10 dataset, we follow the training configuration from Lee et al. Lee et al. (2018). And the training and test result is shown in Table 11.

|  | Training acc. | Test acc. |
|---|---|---|
| ResNet-34 | 100.00±0.00 | 94.84±0.12 |
| DenseNet-100 | 100.00±0.00 | 94.49±0.23 |

Table 11: Classifier training and test accuracy (%) on CIFAR-10.

## H   Generalization of out-of-distribution detection and CIFAR results

As mentioned by Zhou et al. (2021), when exposed to a new different out-distribution dataset, some previous algorithms suffer from severe performance degradation due to a severe bias introduced by validation bias. In the main text, TTACE has shown better average performance on advanced dataset settings like ImageNet. To verify whether our data augmentation and consistency evaluation pipeline has similar bias to out-of-distribution datasets, we conduct CIFAR experiments to test whether there is a generalization problem of TTACE.

The performance on CIFAR-10 settings are shown in Table 12. Specifically, the CIFAR-10 dataset is treated as in-distribution data, while the SVHN Netzer et al. (2011), TinyImageNet Le & Yang (2015), and LSUN Yu et al. (2015) datasets are out-distribution datasets. TTACE reaches the second best results with other algorithms on most settings. At the same time, we should notice that the best algorithm of CIFAR setting in Table 12, i.e. the Mahalanobis, suffers from sever performance degradation in ImageNet settings Table 3, which is consistent with the observations from Zhou et al. (2021). One possible reason is that the features of large network architectures and complicated datasets are not easy to be modeled as Gaussian distributions, which is the main assumption of the Mahalanobis algorithm. Another possible reason is that for ImageNet subset training, the test set size is not as large as CIFAR, so the hyperparameter searching process is difficult on ImageNet settings.

At the same time, our TTACE not only reaches higher average detection performance, but also shows stable generalization ability to low-resolution dataset settings.

| Architecture | Algorithm | Out-distribution data | | |
|---|---|---|---|---|
| | | SVHN | ImageNet_resize | LSUN_resize |
| | MSP | $88.86 \pm 0.96$ | $86.98 \pm 2.01$ | $90.28 \pm 1.13$ |
| | OC-SVM | $90.34 \pm 0.45$ | $\underline{89.64 \pm 0.99}$ | $91.07 \pm 0.50$ |
| ResNet-34 | ODIN | $90.84 \pm 1.71$ | $88.64 \pm 2.77$ | $91.87 \pm 2.31$ |
| | Mahalanobis | $\mathbf{98.50 \pm 0.12}$ | $\mathbf{99.39 \pm 0.06}$ | $\mathbf{99.68 \pm 0.07}$ |
| | TTACE$_{\text{FFT}(t=1)}$ | $\underline{92.95 \pm 0.89}$ | $89.15 \pm 1.93$ | $\underline{92.53 \pm 0.85}$ |
| | TTACE$_{\text{Flip}}$ | $88.59 \pm 0.84$ | $88.01 \pm 1.57$ | $90.67 \pm 0.71$ |
| | MSP | $86.80 \pm 1.64$ | $92.92 \pm 0.87$ | $93.90 \pm 0.66$ |
| | OC-SVM | $77.54 \pm 2.45$ | $80.93 \pm 1.86$ | $82.22 \pm 0.63$ |
| DenseNet-100 | ODIN | $90.76 \pm 0.34$ | $\mathbf{97.73 \pm 0.55}$ | $\mathbf{98.57 \pm 0.34}$ |
| | Mahalanobis | $\mathbf{95.61 \pm 2.23}$ | $91.20 \pm 1.29$ | $80.49 \pm 17.67$ |
| | TTACE$_{\text{FFT}(t=1)}$ | $\underline{91.15 \pm 3.17}$ | $\underline{96.48 \pm 0.94}$ | $\underline{97.88 \pm 0.52}$ |
| | TTACE$_{\text{Flip}}$ | $86.64 \pm 1.62$ | $92.64 \pm 0.77$ | $93.48 \pm 0.56$ |

Table 12: AUROC (%) of CIFAR results. Best results are in **bold**. Second best results are underlined. However, algorithms that reach the best on CIFAR seem to have generalization problems on ImageNet benchmarks. See the main text for more details. TTACE is comparable to others algorithms while maintaining strong generalization ability to other out-of-distribution settings.

# I Sensitivity to filter radius

More sensitivity results for ImageNet subsets are in Figure 11 and Figure 12. In these figures, flatter curves mean less sensitivity to filter radius. For ImageNet subset settings, the optimal FFT filter radius seems to be much lower than that of the full ImageNet setting. A possible explanation is that the frequency domain characteristic of the Artificial dataset is different from the ImageNet dataset because artificial images are drawn by humans which lack fine-grained changes (high frequency signals). Meanwhile, we can see from Table 3 that no matter which radius is used, the performance is still good enough to outperform other algorithms.

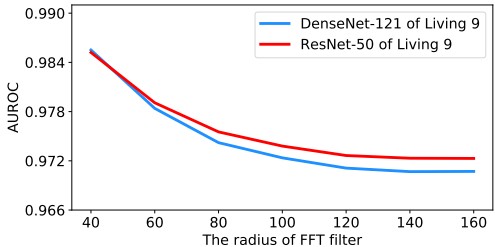 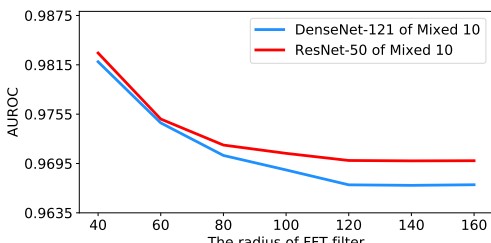

Figure 11: Detection AUROC of Living 9 setting with different radius of FFT filter.

Figure 12: Detection AUROC of Mixed 10 setting with different radius of FFT filter.

# J Sensitivity to temperature

See Figure 13 and Figure 14. Notice that for the *ImageNet vs. Artificial* setting, the dimension of the output is 1000. When the class number is big, temperature effect is enlarged a lot thus large temperature value leads to a relatively fast decreasing in the AUROC result(A brief explanation: Since the maximum output is much bigger than others, when calculating softmax, a large class number will lead to a big denominator value in the softmax equation). And for those whose output dimension is under 20, we can see that the AUROC results are not sensitive to temperature value.

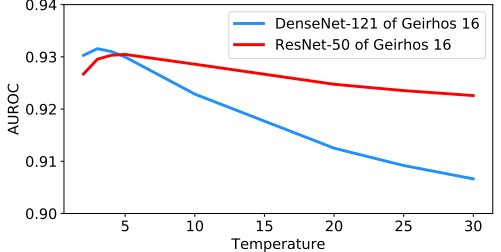 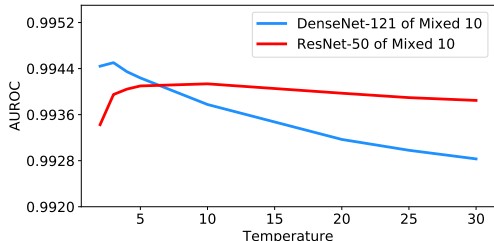

Figure 13: Detection AUROC of Geirhos 16 setting with different temperature.

Figure 14: Detection AUROC of Mixed 10 setting with different temperature.

