# OpenReview forum: "Out-of-distribution Detection with Test Time Augmentation and Consistency Evaluation"
_TMLR — Rejected by TMLR_

### Review · Reviewer_rnvf · 2023-08-04

**Summary Of Contributions:**

The paper proposes an out-of-distribution detection method based on test time augmentation. Both FFT and horizontal flipping are evaluated as augmentation methods. The proposed methods seem to have some marginal improvements, however, the baseline methods compared in the experiments are not sufficiently strong. The proposed method seems only applicable to image-based problems, while out-of-distribution detection is a general problem in machine learning.

**Audience:**

Yes

**Claims And Evidence:**

No

**Requested Changes:**

I recommend a direct rejection to this paper and suggest the authors to submit this work as an extended abstract/poster to a machine learning research workshop.

**Strengths And Weaknesses:**

Strengths:
1. The problem of interest is important to the machine learning community, i.e., out of distribution detection.
2. The proposed method is very simple and easy to implement.
3. The experiments do show some marginal improvements made by the proposal.

Weaknesses:
1. The paper is poorly written and confusing. I do not think its quality meets TMLR standards.
2. It is not reasonable to claim the prior free advantage for the proposed method, given the fact that augmentation methods are pre-defined and chosen by the authors. One could just try a bunch of augmentation methods and cherry-pick the one that optimizes the existing OOD datasets. This is not leading to something generalizable.
3. Test time augmentation is a very common approach. That being said, the novelty of the approach is limited. I suggest the authors to consider a short workshop paper or extended abstract.
4. The experimental validation is weak. First, the baseline methods are old and weak, which are definitely not the state-of-the-art. Second, all the baselines should appear in all comparison tables. Clearly, the paper missed some baselines in some tables. Third, the improvements are mixed, which leads to unclear conclusions.
5. The augmentation methods seem very ad-hoc. I agree they are useful but it does not justify why they are general methods. I would be more interested in a discussion about a good criteria to decide which augmentation methods to use in an input-agnostic situation.
6. The proposed algorithm does not sound. The augmentation function is sampled before hand and fixed through out all data examples. That will lead to high variance in the results.

---

### Review · Reviewer_7Zzm · 2023-09-08

**Summary Of Contributions:**

The authors make a contribution in the detection of out-of-distribution samples (outliers). To do so, the authors use trained networks, and feed to them two views of the same image, but which have different augmentation. Then, they check how different are the predictions of the network, and effectively assign as outliers, the images for which the network give a high mismatch. The authors compare their method with a couple of (relatively old) baselines, showing that their method beats the baselines. Finally, they show some limited ablation studies.

**Audience:**

Yes

**Claims And Evidence:**

No

**Requested Changes:**

I think the authors should improve the paper in the following points, sorted in priority from high to low:

1) Do a more detailed comparison with recent methods. I provided some of them in the review, the authors are encouraged to check for more of them (Point 2 in weaknesses).

2) Do the necessary ablations and try to come with some insights for why the method seems to work reasonably well (Point 3 in weaknesses). The authors are encouraged to do more ablations to increase the paper's value.

3) Do a better job at motivating their method (Point 1 in weaknesses).

4) Improve the writing of the paper and the usage of citations (Point 4 in weaknesses) and try to connect the paper with other domains such as active learning (Point 5 in weaknesses).

**Strengths And Weaknesses:**

I appreciate the following points of the paper:

1) The paper works in a quite important, but underdeveloped field of study.

2) The method seems reasonable, and is quite easy to follow.

3) The experiments shown in the paper seem to beat some other methods.

On the other hand, I think that the paper has the following weaknesses:

1) Lack of clear motivation: the authors motivate their research by doing a simple t-SNE evaluation where they project the features of two datasets and where it can be visually seen that samples of a dataset are far from samples from the other dataset. While this looks like a nice toy example, I think it has the following problems:

a) t-SNE in general is a simple method used to get a bit of feeling for the dataset, but I do not think it can be used for more than that. In fact, even in the t-SNE paper the authors themselves admit that the relatively local nature of t-SNE makes it sensitive to the curse of the intrinsic dimensionality of the data. Obviously, there are also problems with the convergence of the algorithm, so I am not sure that the motivation for this paper can be entirely reached by a simple t-SNE plot.

b) The datasets used for the experiment are CIFAR-10 and SVHN. The datasets are much more different from each other than what would you expect from an outlier coming from the same dataset. I think it might have been much better if they used something that are closer, in order to better emulate the outliers (e.g., some CIFAR-100 images in CIFAR-10 plot).

2) Doubts about the experiments: the method for most part compares only with relatively outdated baselines. Most of them are several years old, and there are more recent methods that significantly outperform those baselines. While the domain does not have a clear experimental setup (for example, domains like tracking, person re-id, metric learning have more established experimental setup), I think the authors should make a better attempt at comparing their method with recent methods in their (recent methods) experimental setup. A non-exhaustive list of these more recent methods:

[a] Deep Hybrid Models for Out-of-Distribution Detection, CVPR 2022

[b] Out-of-Distribution Detection with Semantic Mismatch under Masking, ECCV 2022

[c] Out-of-Distribution Detection via Conditional Kernel Independence Model, NeurIPS 2022

[d] GEN: Pushing the Limits of Softmax-Based Out-of-Distribution Detection, CVPR 2023

3) Not much technical novelty and limited ablations: I hate to write this, but the method does not have much technical novelty. While I do not like to punish papers that are simple, and I agree that often simplicity is the best thing, I also think that in cases where the method is very simple, it needs to give some insights and go beyond we beat the state-of-the-art. On other words, I think there should be better analysis and lessons learned from the paper.

With regards to the ablations, the authors limit the ablations to an analysis of using the remaining classes probability. I wonder how the method would have worked if instead of the original method, or this ablation, there is an ablation on using the entire softmax distribution (for example a Kullback-Leibler, or Jensen-Shannon divergence between the softmax or the first and the second view). In addition, the authors should have ablated using the similarity in feature level instead of probability/softmax level. Furthermore, the authors for augmentations use relatively simple augmentations. I wonder how more sophisticated methods like RandAugment [e] or Trivial Augment [f] would work.

[e] RandAugment: Practical Automated Data Augmentation with a Reduced Search Space, NeurIPS 2020

[f] TrivialAugment: Tuning-free Yet State-of-the-Art Data Augmentation, ICCV 2021

4) The writing/citations could be improved: I think there are some problems with the writing that ideally can be improved in a revised version of the paper:

a) citation Huang et al, 2017 (first line in the introduction) is a bit weird as a citation for deep learning, considering that there were literally tens of thousands of top-tier deep learning papers before that.

b) In the same paragraph the paper mentions a couple of works (Amodei 2016, Goodfellow 2017) that identify some problem, to then mention a paper (Aggarwal 2013) that tries to solve that problem. However, the 'solution' paper came out a few years earlier than the other two papers. I would strongly suggest to the authors to be a bit more careful with the citations, and cite the works accordingly, instead of almost randomly.

c) the third paragraph must be merged with the second paragraph considering that it is talking for the same idea (TTA).

d) the last point in contributions is written in passive.

All these points are in the introduction, with similar points in all parts of the paper. The authors are encouraged to heavily rewrite the paper, make its story more coherent, and be a bit more careful with the citations.

5) Connections with active learning: the problem (and the solution) reminds me quite a lot of active learning problem. There, similarly, the researchers use a network to make some predictions, and then use those predictions to find the most informative samples that need to be labeled. Some of these samples are actually outliers. In particular, the following two methods, are in spirit the same, where they use different views of the same images (using some augmentation method) and then check the softmax distribution disagreement between the two views. So technically, the main difference between those methods and the method presented here is the domain (OOD vs Active Learning).

[g] Consistency-based Semi-supervised Active Learning: Towards Minimizing Labeling Cost, ECCV 2020

[h] Not All Labels Are Equal: Rationalizing The Labeling Costs for Training Object Detection, CVPR 2022

Considering the similarity, I think it would be nice if the authors express the possible connections with those works, and explain the technical differences with them. Furthermore, while this is an OOD paper, it might be interesting to see how it works in the Active Learning setup. It might increase the value of the paper if the authors can show that it works well in a different domain.

---

### Review · Reviewer_qcMB · 2023-09-10

**Summary Of Contributions:**

The paper introduces a post hoc approach for anomaly detection based on evaluating the relationship between the anomaly score and the distance of samples and their transformations when processed by a model 'f.' Specifically, the anomaly score (S(x)) is calculated as 1 − ⟨f(x;t), f(T(x;t))⟩. Experimental results demonstrate the method's effectiveness on benchmark datasets.

**Audience:**

Yes

**Claims And Evidence:**

Yes

**Requested Changes:**

The paper presents a promising post hoc approach for anomaly detection with clear strengths in simplicity and effectiveness. However, it also has notable weaknesses related to its computational cost, robustness to transformations, and limitations in addressing near OOD detection. Further research and experimentation are required to address these concerns and fully assess the method's practical utility.

Addressing Single-Shot Detection:
Action Request: The authors are encouraged to delve deeper into the implications of the proposed single-shot detection approach.  Please explore and discuss potential avenues for making the method more dynamic in handling evolving data distributions.

Robustness and Augmentation Requirement:
Action Request: Consider conducting additional experiments and analyses to investigate the robustness of the method when presented with 'K+1' samples during testing, as opposed to the expected 'K' augmentations. Can the approach be refined to be less sensitive to the number of augmentations? Moreover, explore the possibility of integrating a voting-based mechanism.

Computational Cost:
Action Request: In response to concerns about computational cost, conduct a thorough computational efficiency analysis. Are there optimizations or techniques that can be employed to reduce the method's computational overhead without compromising its performance? Provide insights into how the method can be made more computationally efficient.

Handling Data Invariance to Transformations:
Action Request: Investigate strategies to enhance the method's resilience when dealing with data samples that exhibit invariance to specific transformations. Can additional features or pre-processing steps be incorporated to address this limitation? Share insights into how the method can better handle scenarios where transformations have minimal impact.

Near Out-of-Distribution (OOD) Detection:
Action Request: Extend the scope of the research to include near OOD detection scenarios. Perform experiments and evaluations to assess how the method performs when data samples are in proximity to, but not strictly within, the learned distribution. Provide a comprehensive analysis of the method's performance in these scenarios.

**Strengths And Weaknesses:**

Strengths:

Clarity of Presentation: The paper is commendably well-written and easily comprehensible. It effectively conveys the proposed approach and its rationale.

Simplicity and Effectiveness: The method's simplicity is a notable strength. Despite its simplicity, it proves to be highly effective for anomaly detection, which is a valuable contribution to the field.

Interesting Insight: The finding that there exists a distinct relationship between the distance of samples and their augmentations for normal and abnormal samples is intriguing. This insight adds depth to the research.

Weaknesses:

Detecting anomalies usually requires evaluating whether a sample is consistent with the pattern learned by the model. The suggested approach entails feeding the model with multiple times of the sample and its augmentation, and then determining if the input is normal or not.

To test a sample 'X', the 'Augmentation Requirement' method requires 'K' augmentations. However, if 'K+1' samples are fed to the model 'f' and analyzed by an MSP anomaly detector and a voting-based approach, the effectiveness of the method is questioned. Does MSP+voting offer the same level of performance?

Computational Cost: The paper acknowledges computational cost as a concern, but it remains a significant drawback that needs further attention.

Invariance to Transformations: The method may fail when dealing with samples that are invariant to certain transformations. For example, if 'X' is an out-of-distribution image of a chessboard, the method could potentially produce erroneous results when the image is flipped or rotated.

Limited Application to Near OOD Detection: The paper does not address near out-of-distribution (OOD) detection, which is a crucial aspect of anomaly detection. Evaluating the method's performance in scenarios where samples are close to the learned distribution should be considered.

---

### Decision · Action_Editor_iwXv · 2023-11-13

**Recommendation:** Reject

**Comment:**

There is a unanimous recommendation from the reviewers on rejection of the paper. Some common main concerns include the limited technical contribution of the proposed work and the lack of solidness in the experiments/evaluation. The authors have also not addressed the wide concerns from reviewers. Test-time-augmentation is a commonly used technique in transfer learning. Recently, the vision-language community has also adopted similar techniques for applications such as prompt tuning. The paper, in its current form, has clearly under-represented these relevant vast progress.

**Audience:**

Some audience in the transfer learning community may be interested in the findings in this paper.

**Claims And Evidence:**

This paper proposes a test-time-augmentation and consistency evaluation based approach to address out-of-distribution detection in a post-hoc manner. The proposed method is very simple and straightforward. While being effective, the work lacks enough technical contributions and have not sufficiently addressed concerns/weaknesses regarding the approach. The evaluation and experiment of this work overall do not seem solid.